# Diagnosis of bone metastases in breast cancer: Lesion-based sensitivity of dual-time-point FDG-PET/CT compared to low-dose CT and bone scintigraphy

Jeanette Ansholm Hansen[1,2], Mohammad Naghavi-Behzad[1,3]*, Oke Gerke[1,3], Christina Baun[1,3], Kirsten Falch[1], Sandra Duvnjak[4,5], Abass Alavi[6], Poul Flemming Høilund-Carlsen[1,3], Malene Grubbe Hildebrandt[1,3,7,8]

1 Department of Clinical Research, University of Southern Denmark, Odense, Denmark, 2 Department of Obstetrics and Gynecology, Odense University Hospital, Odense, Denmark, 3 Department of Nuclear Medicine, Odense University Hospital, Odense, Denmark, 4 Radiology Department–Breast Imaging, Herlev Gentofte Hospital, Copenhagen, Denmark, 5 Mammography Screening Center in the Capital Region, Herlev Gentofte Hospital, Copenhagen, Denmark, 6 Division of Nuclear Medicine, Department of Radiology, University of Pennsylvania, Perelman School of Medicine, Philadelphia, PA, United States of America, 7 Centre for Innovative Medical Technology, Odense University Hospital, Odense, Denmark, 8 Centre for Personalized Response Monitoring in Oncology, Odense University Hospital, Odense, Denmark

* Mnb91@rsyd.dk

**Data Availability Statement:** All relevant data are within the paper and its Supporting Information files.

## Abstract

We compared lesion-based sensitivity of dual-time-point FDG-PET/CT, bone scintigraphy (BS), and low-dose CT (LDCT) for detection of various types of bone metastases in patients with metastatic breast cancer. Prospectively, we included 18 patients with recurrent breast cancer who underwent dual-time-point FDG-PET/CT with LDCT and BS within a median time interval of three days. A total of 488 bone lesions were detected on any of the modalities and were categorized by the LDCT into osteolytic, osteosclerotic, mixed morphologic, and CT-negative lesions. Lesion-based sensitivity was 98.2% (95.4–99.3) and 98.8% (96.8–99.5) for early and delayed FDG-PET/CT, respectively, compared with 79.9% (51.1–93.8) for LDCT, 76.0% (36.3–94.6) for BS, and 98.6% (95.4–99.6) for the combined BS+LDCT. BS detected only 51.2% of osteolytic lesions which was significantly lower than other metastatic types. SUVs were significantly higher for all lesion types on delayed scans than on early scans (P<0.0001). Osteolytic and mixed-type lesions had higher SUVs than osteosclerotic and CT-negative metastases at both time-points. FDG-PET/CT had significantly higher lesion-based sensitivity than LDCT and BS, while a combination of the two yielded sensitivity comparable to that of FDG-PET/CT. Therefore, FDG-PET/CT could be considered as a sensitive one-stop-shop in case of clinical suspicion of bone metastases in breast cancer patients.

**Funding:** This research was partially supported by the Independent Research Fund Denmark (DFF-7016-00359) and the Centre of Personalized Response Monitoring in Oncology at Odense University Hospital (Denmark). The funders had no role in study design, data collection and analysis, decision to publish, or preparation of the manuscript.

**Competing interests:** The authors have declared that no competing interests exist.

## Introduction

Breast cancer mortality is almost exclusively a result of distant metastatic disease [1] with survival rates of 99% for patients with localized disease and only 25% for patients with metastatic disease [2]. Bone is the most common site of metastasis in patients with breast cancer, occurring in up to 70% of patients with advanced disease [1,3]. This leads to chronic metastatic bone disease in many, since relevant treatment can often delay progression [4]. Bone metastases seem to originate in bone marrow, and structural changes in the bone will occur in a postponed phase [5]. Structural changes in the bone can be detected and classified as osteolytic, osteosclerotic, or mixed (osteolytic/osteosclerotic) metastases [6]. Planar bone scintigraphy (BS) reflects osteoblastic activity and is probably superior in detecting osteosclerotic and mixed metastases than other types of bone metastases [7]. BS and computed tomography (CT) are the most often used modalities of conventional imaging and are recommended in current guidelines for the detection of bone metastases in breast cancer [8,9].

[18F]-fluorodeoxyglucose-Positron Emission Tomography with integrated computed-tomography (FDG-PET/CT) reflects glucose metabolism, and thus this modality may facilitate detection of all types of bone metastases including bone marrow metastases [10]. It is well known that breast cancer patients represent various types of bone metastatic lesions, but the literature is equivocal regarding whether FDG-PET/CT or conventional imaging is superior in detection of bone metastases [11–13]. However, it has often been pointed out that FDG-PET/CT may be superior in detecting osteolytic rather than osteosclerotic bone lesions in breast cancer patients [13–17].

The ability to distinguish malignant from benign lesions with FDG-PET/CT may be improved by delayed imaging [18,19]. Also, delayed imaging using FDG may therefore be particularly useful for diagnosing low metabolic malignancies such as breast cancer and especially for the less detectable lesions on regular FDG-PET/CT such as osteosclerotic bone metastases [19].

Considering the fact that bone involvement is the predilection site for metastasis in breast cancer patients, and when the progression of bone metastasis is not detected and taken care of, the risk of developing skeletal-related events increases and result in higher risk of mortality [20,21]. We hypothesized that delayed FDG-PET/CT scan would act more accurately regarding the detection of bone metastatic lesions than on early FDG-PET/CT and conventional imaging. Therefore, we aimed to investigate the lesion-based sensitivity of dual-time-point FDG-PET/CT compared with BS and low-dose CT (LDCT) for the detection of bone metastases in breast cancer patients. Furthermore, we aimed to determine FDG standardized uptake values (SUVs) in different types of bone lesions at early and delayed images; however the small sample size is a critical limitation to this specific aim.

## Materials and methods

### Study design and subjects

This prospective study was carried out at the Department of Nuclear Medicine of Odense University Hospital (Odense, Denmark). A written informed consent form was obtained from all included patients and the study protocol was approved by the ethics committee (S-20110138) at the University of Southern Denmark (Odense, Denmark), which was in compliance with good clinical practice and the Declaration of Helsinki (Registration code at ClinicalTrials.gov: NCT01552655).

In a prospective comparative design, patients with suspected breast cancer recurrence or with verified local recurrence and potential distant disease, referred from the Department of

Oncology between 2011 (Dec) and 2014 (Sep), were considered eligible for the inclusion. Exclusion criteria were history of concurrent malignancy, age younger than 18 years, pregnancy or breast-feeding, diagnosed diabetes mellitus, or considered unable to cooperate. All of the patients who accepted participation were asked to undergo dual-time-point FDG-PET/CT and whole-body BS, within a median time interval of three days (range: 0–24). The patients with histopathologically confirmed metastatic breast cancer with approved bone involvement were included in analysis. All patients initiated systemic therapy based on the biopsy-verified diagnosis and according to national oncologic guidelines for metastatic breast cancer [22]. Overall patient-based accuracy results of this study have been published previously [23], and the current analysis considered lesion-based sensitivity focusing on various types of bone metastases along with respective quantification measures reflecting FDG-uptake.

### FDG-PET/CT protocol

Before the FDG-PET/CT scan, patients were required to fast for at least 6 h, after which their blood sugar levels were measured. PET/CT was considered acceptable at levels up to 144 mg/dL. The 18F-FDG tracer was administered intravenously with an activity of 4 MBq per kg of body weight. The patients were requested to rest for 60 min (±5 min) p.i. before PET/CT imaging was performed from the top of the skull to the proximal femur [24]. The second scan was performed in the same manner after 180 min (±5 min) [25]. The total examination time was approximately 210 min for each patient. All scans were performed using either the Discovery STE (VCT) equipped with BGO crystals or the Discovery RX equipped with LYSO(Ce) crystals (GE Healthcare Systems, Chicago, IL, USA). PET was performed over 7–9 bed positions in 3D, with a scan time of 2.5 min per bed position for 1-h images and 3.5 min per bed position for 3-h images. PET images were reconstructed iteratively, with ordered subset expectation maximization, 2 iterations, and 21 or 28 subsets.

### LDCT protocol

Low-dose CT imaging, with two scout views for both exams, was performed using either GE Discovery STE or Discovery RX (GE Medical Systems, Milwaukee, WI), at 140 kV with SmartmA tube current modulation (noise index 35, 0.8 seconds per rotation, slice thickness 3.75 mm) and used for attenuation correction and anatomic orientation followed by a 3D PET scan (OSEM iterative reconstruction, slice thickness 3.75 mm) [26].

### Bone scintigraphy

The patients were injected with 700 MBq (0.019 Ci) Technetium-99m-3,3-disphosphono-1,2-propanodicarboxylic acid (Tc-99m-DPD) three to four hours prior to whole-body imaging. In the waiting period, the patients were asked to drink approximately 1 liter of clear liquids. The scan was performed on a Skylight or PRISM XP2000 gamma camera (Philips Medical, Surrey, UK) with the following parameters: LEHR collimator, energy window 140 keV ± 20%, matrix 256 x 1024, scan speed 14 cm/min.

### Reference standard

Suspected recurrence was verified by biopsy as the reference standard. All patients treated explicitly for bone metastasis, typically with bisphosphonates, were categorized as having bone metastases. Follow-up time was defined as the time interval between the date of the first scan and the date of the latest registered clinical contact to the Departments of Clinical Oncology or Breast Surgery.

## Image interpretation

All FDG-positive bone lesions present on 1h or 3h FDG-PET/CT scans were counted by single group of nuclear medicine specialists through daily practice. BS studies were examined to identify the FDG-positive lesions and potential additional lesions. An experienced radiologist categorized metastatic bone lesions into osteolytic (partially ill-defined margin with pattern of bone resorption and focal bone destruction), osteosclerotic (dense and often well-defined margin with pattern of bone formation and ossification), and mixed subtypes based on radiographic features of the LDCT. FDG-positive lesions without changes on LDCT were designated as "CT-negative metastases" [27,28]. All bone lesions were categorized as positive in patients with confluent FDG-uptake in bone on FDG-PET/CT or confluent Tc-99m-DPD uptake on BS (super scan). All bone lesions detected by FDG-PET/CT, LDCT, or BS were considered positive, although degenerative lesions in large joints were not included. The radiologist had the LDCT and BS scans in two separate screens (side by side) for the lesion categorization through LDCT+CT.

## Lesion-based sensitivity and quantification

Lesion-based sensitivity with 95% confidence intervals (95% CIs) was calculated for all three modalities and for the combined LDCT+BS. FDG-avid bone lesions were quantified using dedicated software (ROVER, ABX, Radeberg, Germany) to determine maximum and mean SUVs and the latter corrected for partial volume effect (SUVmax, SUVmean, cSUVmean). Segmentation of bone lesions was obtained by manually placing a three-dimensional mask on all suspected lesions and delineating the region of interest (ROI) by using a threshold of 40% of the maximum value of the three-dimensional mask. We included a minimum ROI volume of one cubic centimeter and excluded ROI intersections [29]. The software then automatically calculated metabolically active volume (MAV) for each ROI [30]. The retention index of each lesion was calculated as follows [19]:

$$\text{Retention Index} = (\text{SUV[3h]} - \text{SUV[1h]})/\text{SUV[1h]} \times 100\%$$

## Statistical analyses

Descriptive statistics were performed according to the data type (continuous: median and range; categorical: frequencies and percentages). Simple linear regression was used to test for differences in SUVs and MAV between different bone lesion types with 3h and 1h FDG-PET/CT imaging. Clustered sandwich estimators were used in both linear regression and derivation of 95% CIs to account for clustered data. P-values of $<0.05$ were considered significant. All statistical analyses were conducted with STATA/MP 16 (StataCorp, College Station, 77845 Texas, USA).

# Results

## Demographic information

Eighteen patients with a median age of 61.5 years (range: 38–76) had confirmed bone recurrence; 7 by bone biopsies and 11 by biopsies from other sites with confirmation of bone involvement by further imaging, or retrospectively observed progression in bone lesions on later scans. The patients were followed-up for a median period of 19 months (range: 1–35 months). Baseline characteristics of included patients are summarized in Table 1.

**Table 1. Baseline characteristics of included patients with metastatic breast cancer.**

| Variable | | Results* | Variable | | Results* |
|---|---|---|---|---|---|
| **Primary tumor size** (mm) | | 21 (10–70) | **Estrogen receptor status** | Positive | 15 (83.3) |
| **Time until relapse**** (month) | | 60 (0–324) | | Negative | 2 (11.1) |
| **Histopathology** | Invasive ductal carcinoma | 15 (83.3) | | Unknown | 1 (5.6) |
| | Invasive lobular carcinoma | 3 (16.7) | **Herceptin-2 receptor status** | Positive | 3 (16.7) |
| **Surgery type** | Lumpectomy | 7 (38.9) | | Negative | 14 (77.8) |
| | Mastectomy | 11 (61.1) | | Unknown | 1 (5.6) |
| **Treatment protocol** | Chemotherapy | 13 (72.2) | **Malignancy Grade** | 1 | 3 (16.7) |
| | Hormone therapy | 12 (66.7) | | 2 | 7 (38.9) |
| | Radiotherapy | 15 (83.3) | | 3 | 8 (34.4) |

*Data are shown as frequency (%) and median (interquartile range).

**Time period between primary breast cancer and diagnosis of metastasis.

## Lesion-based sensitivity

A total of 488 bone lesions were detected by any modality with a median of five lesions per patient (range: 1–99). Three FDG-PET/CT studies did not include the head by technical mistake. Four patients had a super scan on BS, and seven patients (39.9%) had more than ten bone lesions. FDG-PET did not identify five osteolytic skull lesions, four of which were detected by both LDCT and BS, one by LDCT only.

The lesion-based sensitivity for each modality is presented in Table 2. Early and delayed FDG-PET/CT images had higher sensitivity compared with BS and LDCT separately, while they showed almost the same sensitivity when compared with the combined BS+LDCT. Sixty-two of 98 (63%) CT-negative lesions on LDCT were located in the ribs, humerus, scapula, or clavicles.

BS detected significantly fewer osteolytic lesions (104/213) than other bone metastatic lesions (267/275). Also, BS could not identify any lesion in three patients and detected only a few of several lesions (1/17 and 8/87) in two patients (Figs 1 and 2).

One patient with bone metastatic lobular carcinoma presented with diffuse osteosclerotic changes in the skeleton that did not take up FDG. The diffuse appearance made the lesions uncountable. She had seven lytic lesions that were FDG-avid and therefore counted as true positive on FDG-PET/CT.

**Table 2. Types of detected lesions and lesion-based sensitivity by each modality.**

| Detected lesions / Modality | Lesion type | | | | | Lesion-based sensitivity(95% CI) |
|---|---|---|---|---|---|---|
| | Osteolytic | Osteosclerotic | Mixed | CT-negative | All lesions | |
| **LDCT** | 213 | 80 | 97 | 0 | 390 | 79.9 (51.1–93.8) |
| **BS** | 104 | 79 | 97 | 91 | 371 | 76.0 (36.3–94.6) |
| **FDG-PET/CT (1h)** | 206 | 78 | 97 | 98 | 479 | 98.2 (95.4–99.3) |
| **FDG-PET/CT (3h)** | 208 | 79 | 97 | 98 | 482 | 98.8 (96.8–99.5) |
| **LDCT+ BS** | 213 | 80 | 97 | 91 | 481 | 98.6 (95.4–99.6) |

CI: Confidence interval; LDCT: Low-dose computed tomography; BS: Bone scintigraphy; FDG-PET/CT, Fluorodeoxyglucose positron emission tomography with integrated computed-tomography.

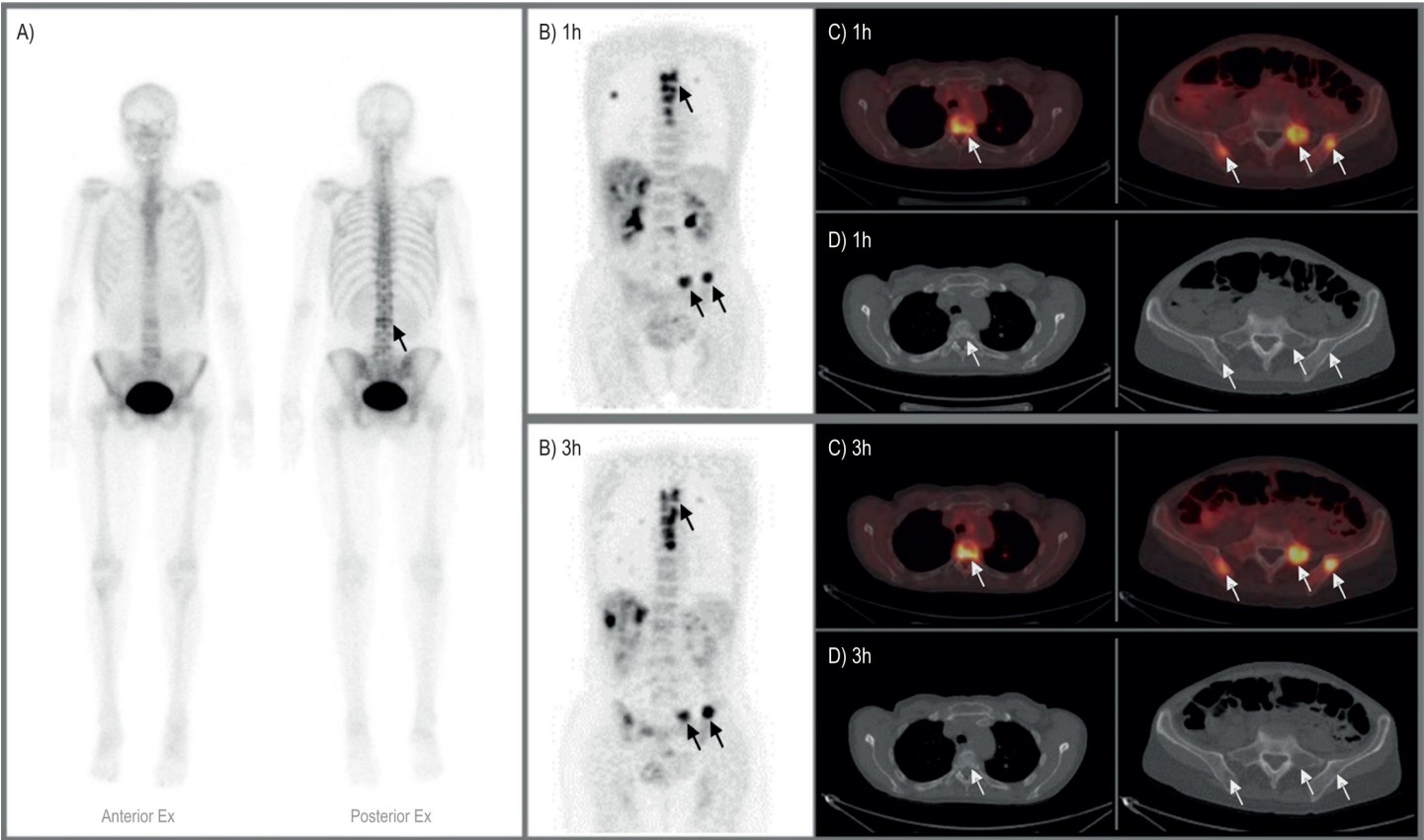

**Fig 1. A 54-year-old woman with true-positive bone metastases.** A) Whole-body bone scintigraphy shows only one area with increased uptake of 99mTc-DPD (arrow). B) FDG-PET 1h and 3h images show multiple osseous metastases in the spine and the pelvis. C) Axial FDG-PET/CT images demonstrating FDG-avid lesions in the spine, sacrum, and iliac bones. D) Axial CT images at the same level as C show osteolytic changes.

### Quantification findings

Seven lesions were located in the skull and were excluded from quantification analyses due to scatter from high FDG-uptake in the cerebrum. The remaining 481 lesions showed a statistically significant 1h to 3h increase in SUVmax, SUVmean, and cSUVmean for all lesion types (P<0.0001, Table 3). Osteolytic and mixed-type lesions had higher SUVs than osteosclerotic and CT-negative metastases at both time-points. The 1h to 3h increase in SUVs was lower for osteosclerotic than other lesion types. The median retention index was significantly lower in osteosclerotic lesions compared with other types of lesions (P = 0.006). Comparison of early and delayed cSUVmean through different lesion types is shown in Fig 3.

### Discussion

FDG-PET/CT was superior to BS and LDCT regarding the detection of bone metastases in patients with recurrent metastatic breast cancer. This modality had significantly higher lesion-based sensitivity for bone recurrence than LDCT or BS alone, in particular, because it was much better than BS for the detection of osteolytic lesions and superior to LDCT in the detection of lesions which were deemed invisible by LDCT (CT-negative metastases). Early and delayed FDG-PET/CT images showed almost the same sensitivity (98.2% vs. 98.8%). Although all types of bone metastases showed increased FDG-uptake and were equally detectable at 1h

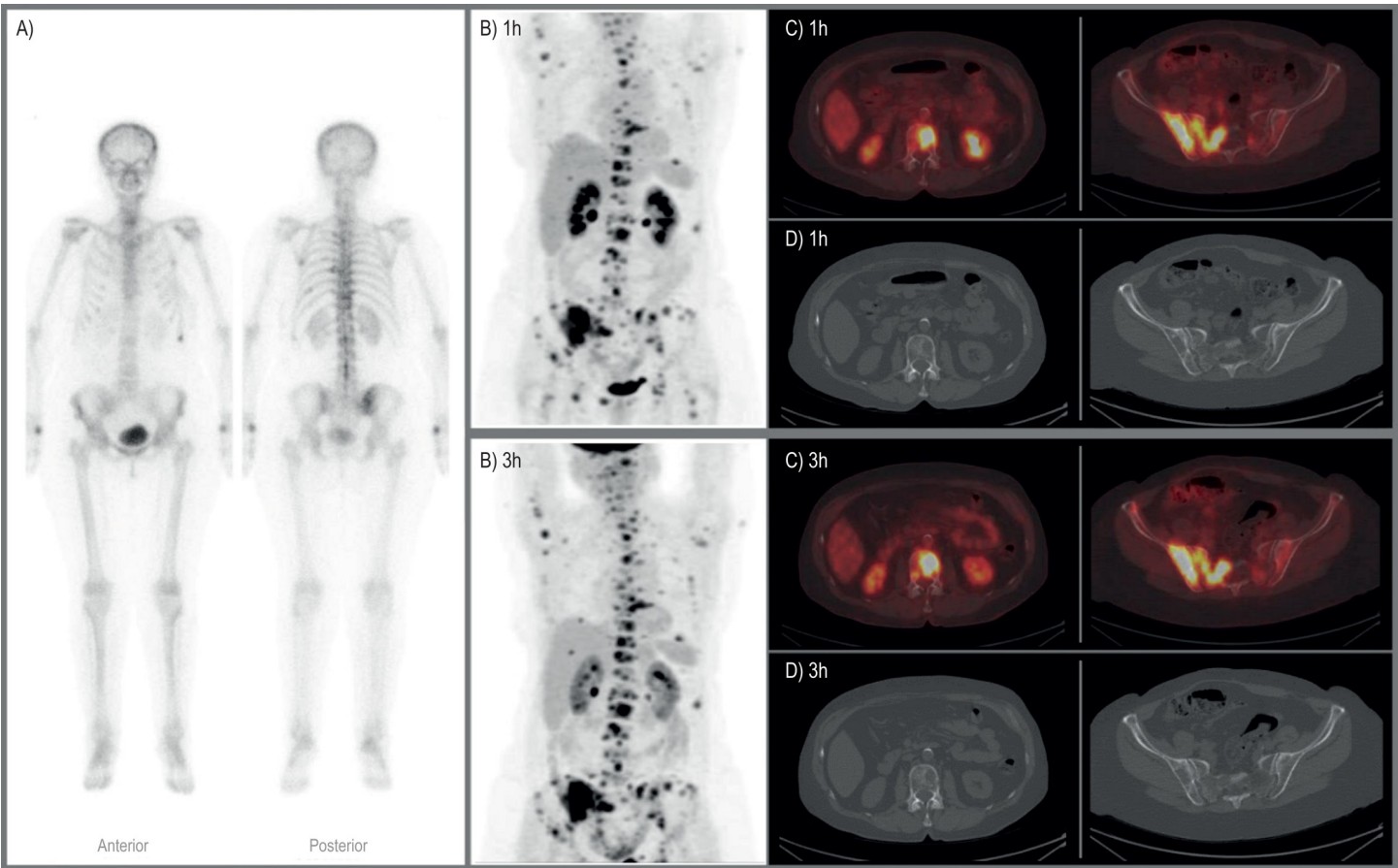

**Fig 2. A 71-year-old woman with true-positive bone metastases.** A) Whole-body bone scintigraphy shows few areas with increased uptake of 99mTc-DPD osteolytic lesions. B) FDG-PET images show multiple osseous metastases in the skeleton and metastases in other organs on 1h and 3h images. C) Axial 1h and 3h FDG-PET/CT images showing FDG-avid lesions in the spine, sacrum, and iliac bones. D) Axial CT images at the same level as C show osteolytic changes.

and 3h images and with high respective lesion-based sensitivities, FDG-PET/CT parameters (SUVs and retention index) were significantly lower in osteosclerotic lesions compared with the others.

Strengths of our study were the prospective design, which all patients were treatment-naive concerning bone metastases, and that patients acted as their own controls during the follow-up time. Besides, the short time interval between imaging procedures, using of the experienced readers for each specific modality, the implication of dedicated software for PET quantification, and LDCT of the same field of view as with FDG-PET/CT could count as the advantages of the current study. Limitations were a relatively small sample size with a skewed range of lesions per patient, that image modalities could not be blinded for the lesion-based analysis that osteosclerotic lesions were more difficult to characterize due to their more diffuse appearance and that only a single biopsy from each patient dictated the origin of the majority of lesions. Furthermore, BS was without SPECT/CT and that 18F-Sodium Fluoride PET/CT and contrast-enhanced CT were not included in the comparison.

In a retrospective Japanese study of 88 breast cancer patients with bone metastasis, they found higher lesion-based sensitivity (94%) for FDG-PET/CT than for CT and BS (77% and 89%, respectively), which were in line with the results of our study. However, they found a relatively lower detection of osteosclerotic lesions for FDG-PET/CT than other lesion types [13],

**Table 3. FDG uptake and metabolically active volume in types of bone metastasis[*].**

| Quantitative measure / Lesion type | | Osteolytic (n = 207) | Osteosclerotic (n = 79) | Mixed (n = 97) | CT-negative (n = 98) | All lesions (n = 481) |
|---|---|---|---|---|---|---|
| SUVmax | 1h | 6.0 (1.2–16.6) | 4.4 (1.5–11.8) | 6.5 (3.1–14.9) | 3.8 (1.7–11.5) | 5.3 (1.2–16.6) |
| | 3h | 7.7 (1.8–21.2) | 5.5 (2.3–15.5) | 8.4 (2.7–21.1) | 5.1 (2.1–14.4) | 6.6 (1.8–21.2) |
| | Δ | 1.5 (-1.4–7.1) | 0.9 (-0.6–5.2) | 2.2 (-4.8–7.3) | 1.2 (-0.8–5.0) | 1.4 (-4.8–7.3) |
| SUVmean | 1h | 4.0 (0.9–10.0) | 3.1 (1.1–6.3) | 4.2 (1.8–9.2) | 2.6 (1.0–7.4) | 3.6 (0.9–10.0) |
| | 3h | 5.1 (1.0–13.1) | 3.8 (1.4–8.1) | 5.1 (2.0–11.7) | 3.1 (1.3–9.6) | 4.5 (1.0–13.1) |
| | Δ | 0.9 (-0.9–4.7) | 0.6 (-0.6–2.6) | 1.2 (-2.7–4.1) | 0.7 (-0.2–4.6) | 0.9 (-2.7–4.7) |
| cSUVmean | 1h | 7.6 (0.9–36.4) | 5.2 (1.6–17.0) | 7.5 (2.1–19.9) | 5.0 (1.2–15.9) | 6.7 (0.9–36.4) |
| | 3h | 10.2 (1.1–26.0) | 6.3 (2.0–25.8) | 10.0 (2.9–36.3) | 5.9 (1.7–19.1) | 4.5 (1.0–13.1) |
| | Δ | 2.1 (-18.5–14.1) | 1.2 (-5.9–15.5) | 3.0 (-12.7–21.7) | 1.2 (-10.5–11.9) | 1.8 (-18.5–21.7) |
| Metabolically active volume (cm³) | 1h | 1.8 (0.1–65.1) | 4.2 (0.2–31.0) | 3.6 (0.4–61.9) | 2.4 (0.3–26.6) | 2.5 (0.1–65.1) |
| | 3h | 1.9 (0.2–71.7) | 3.9 (0.3–35.5) | 3.4 (0.4–52.3) | 2.0 (0.5–21.7) | 2.3 (0.2–71.7) |
| | Δ | -0.1 (-34.6–6.6) | 0.0 (-5.6–8.3) | -0.4 (-18.2–4.3) | -0.1 (-14.5–2.8) | -0.1 (-34.6–8.3) |
| Retention index (%) | | 25.0 (-28.0–125.0) | 20.0 (-14.6–81.0) | 34.5 (-57.1.102.1) | 29.4 (-13.6–166.7) | 27.7 (-57.1–166.7) |

SUV: Standardized uptake value; cSUVmean: Corrected SUVmean.

[*]Data was shown as median (interquartile range).

which was not confirmed by our study. Additional to the results of previous studies regarding the superiority of FDG-PET/CT in detection of bone metastases compared to BS [11,12,15], our study showed that LDCT and BS combined could provide sensitivity equal to that of FDG-PET/CT in detection of skeletal metastases.

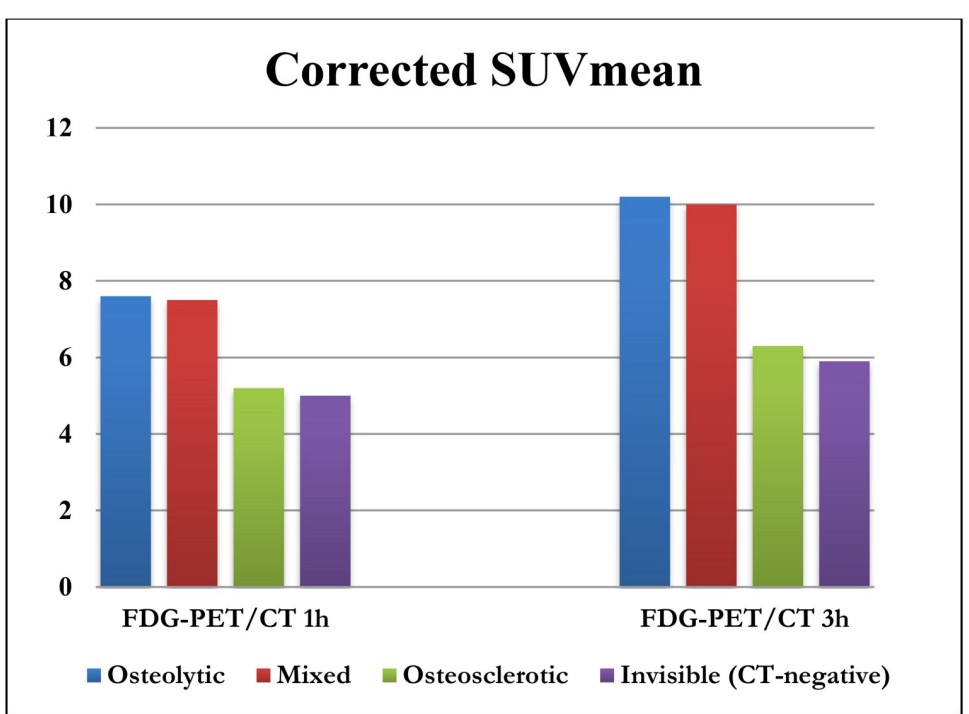

**Fig 3. Comparison of early and delayed corrected standardized uptake value for partial volume within different lesion types (FDG-PET/CT: Fluorodeoxyglucose positron emission tomography with integrated computed-tomography; Corrected-SUVmean: Corrected standardized uptake value for partial volume).**

FDG-PET/CT has also previously been reported to be superior to BS in the detection of osteolytic and less sensitive in detecting osteosclerotic lesions [13,14]. However, in retrospective studies, it may be likely that some patients are not treatment-naive, and in that case, FDG-negative osteosclerotic lesions may represent bone healing. We detected FDG-positive osteosclerotic lesions although we did find higher FDG-uptake in osteolytic and mixed metastases than in osteosclerotic and CT-negative lesions, thus supporting previous findings to some degree. The delayed FDG-PET/CT images had in general better tumor-to-background discrimination and improved image quality in agreement with previous reports on delayed imaging [31]. Nonetheless, the improved image quality and higher SUVs at delayed scans did not translate into significantly higher detection sensitivity.

In a recently published paper, comparing clinical management of metastatic breast cancer patients undergoing BS, contrast-enhanced CT, and FDG-PET/CT regarding the assessment of bone metastasis, it has been shown that FDG-PET/CT resulted in clinically relevant management differences in 16% of patients compared with BS [32]. Since it has already been approved that early detection of bone metastasis plays an important role in the survival of patients with metastatic breast cancer [33], the clinical application of FDG-PET/CT may guide the treatment better than when using conventional imaging [34,35].

Therefore, proper detection of bone metastases is crucial for the choice of proper treatment. Previous studies showed higher patient-based sensitivities with FDG-PET/CT than with conventional imaging when diagnosing bone recurrence [23,36]. These findings suggest that oligometastatic bone disease can be detected earlier by FDG-PET/CT than by conventional imaging. Also, patient-based specificity was improved by FDG-PET/CT, which may significantly benefit patients and reduce management costs in this particular patient group.

Our results indicated that FDG-PET/CT, compared with conventional imaging, could act more sensitive regarding the detection of bone metastasis and distinguishing the different types of bone lesions. However, these results need to be approved by prospective larger studies which include 18F-Sodium Fluoride PET/CT and contrast-enhanced CT to the comparison in order to achieve a firm conclusion about the most sensitive modality to detect bone metastasis. Additional information derived from follow-up scans could provide relevant results on diagnostic accuracy of FDG-PET/CT in response evaluation of skeletal metastases and needs to be considered in future studies.

## Conclusions

FDG-PET/CT had significantly higher lesion-based sensitivity than low-dose CT or bone scintigraphy alone and thus, may act more clinically useful as a one-stop-shop for diagnosing bone recurrence in breast cancer patients. FDG-PET/CT had significantly higher sensitivity than BS and LDCT for the detection of osteolytic metastases and lesions appearing in the bone marrow, respectively. Delayed FDG-PET/CT imaging did not improve lesion-based sensitivity significantly.

## Supporting information

**S1 Data. Data bone.**
(XLSX)

## Author Contributions

**Conceptualization:** Jeanette Ansholm Hansen, Mohammad Naghavi-Behzad, Oke Gerke, Christina Baun, Sandra Duvnjak, Abass Alavi, Poul Flemming Høilund-Carlsen, Malene Grubbe Hildebrandt.

**Data curation:** Jeanette Ansholm Hansen, Mohammad Naghavi-Behzad, Oke Gerke, Christina Baun, Kirsten Falch, Sandra Duvnjak, Malene Grubbe Hildebrandt.

**Formal analysis:** Jeanette Ansholm Hansen, Oke Gerke.

**Funding acquisition:** Christina Baun, Poul Flemming Høilund-Carlsen, Malene Grubbe Hildebrandt.

**Investigation:** Mohammad Naghavi-Behzad, Oke Gerke, Christina Baun, Kirsten Falch, Sandra Duvnjak.

**Methodology:** Oke Gerke, Christina Baun, Kirsten Falch, Sandra Duvnjak, Abass Alavi, Poul Flemming Høilund-Carlsen, Malene Grubbe Hildebrandt.

**Project administration:** Mohammad Naghavi-Behzad, Christina Baun, Malene Grubbe Hildebrandt.

**Software:** Christina Baun, Kirsten Falch.

**Supervision:** Oke Gerke, Sandra Duvnjak, Abass Alavi, Poul Flemming Høilund-Carlsen, Malene Grubbe Hildebrandt.

**Validation:** Oke Gerke, Christina Baun.

**Visualization:** Christina Baun, Sandra Duvnjak.

**Writing – original draft:** Jeanette Ansholm Hansen, Mohammad Naghavi-Behzad, Oke Gerke, Christina Baun, Kirsten Falch, Sandra Duvnjak, Abass Alavi, Poul Flemming Høilund-Carlsen, Malene Grubbe Hildebrandt.

**Writing – review & editing:** Jeanette Ansholm Hansen, Mohammad Naghavi-Behzad, Oke Gerke, Christina Baun, Kirsten Falch, Sandra Duvnjak, Abass Alavi, Poul Flemming Høilund-Carlsen, Malene Grubbe Hildebrandt.

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
