## [Decision Letter · Decision Letter 0]

12 Jul 2021

PONE-D-20-39838

Diagnosis of Bone Metastases in Breast Cancer: Lesion-Based Sensitivity of Dual-Time-Point FDG-PET/CT Compared to Low Dose CT and Bone Scintigraphy

PLOS ONE

Dear Dr. Naghavi-Behzad,

Thank you for submitting your manuscript to PLOS ONE. After careful consideration, we feel that it has merit but does not fully meet PLOS ONE’s publication criteria as it currently stands. Therefore, we invite you to submit a revised version of the manuscript that addresses the points raised during the review process.Please submit your revised manuscript by Aug 21 2021 11:59PM. If you will need more time than this to complete your revisions, please reply to this message or contact the journal office at plosone@plos.org. Please include the following items when submitting your revised manuscript:

We look forward to receiving your revised manuscript.

Kind regards,

Matteo Bauckneht

Academic Editor

PLOS ONE

Journal Requirements:

2. Please provide additional details regarding participant consent. In the ethics statement in the Methods and online submission information, please ensure that you have specified whether consent was informed.

This study was supported by University of Southern Denmark and Odense University Hospital (Odense, Denmark).

This study was supported by University of Southern Denmark and Odense University Hospital (Odense, Denmark).

6. Please upload a new copy of Figures 1 and 2 as the detail is not clear. Please follow the link for more information: https://blogs.plos.org/plos/2019/06/looking-good-tips-for-creating-your-plos-figures-graphics/" https://blogs.plos.org/plos/2019/06/looking-good-tips-for-creating-your-plos-figures-graphics/ .

Reviewers' comments:

Reviewer's Responses to Questions

**Comments to the Author**

1. Is the manuscript technically sound, and do the data support the conclusions?

Reviewer #1: Partly

Reviewer #2: Yes

Reviewer #3: Partly

2. Has the statistical analysis been performed appropriately and rigorously? 

Reviewer #1: Yes

Reviewer #2: Yes

Reviewer #3: I Don't Know

3. Have the authors made all data underlying the findings in their manuscript fully available?

Reviewer #1: Yes

Reviewer #2: Yes

Reviewer #3: Yes

4. Is the manuscript presented in an intelligible fashion and written in standard English?

Reviewer #1: Yes

Reviewer #2: Yes

Reviewer #3: Yes

5. Review Comments to the Author

Reviewer #1: Major comments:

This manuscript is well-written, however, there are some previous studies have reported the similar results and conclusions, such as Acta Radiol, 2011, 52(9):1009-14., J Clin Oncol,2016, 34(16): 1889-97., Mol Imaging Biol, 2012, 14(2): 252-9., and Mol Imaging Biol, 2020, 22(2): 397-406.

Minor comments:

1. Please write down the inclusion and exclusion criteria of samples in detail in Study Design and Subjects.

2. The scanning range of PET-CT should be performed from the top of the skull to the proximal femur, and metastatic lesions of the skull should not be excluded in the study.

3. Is there any other literature supporting the definition of "invisible metastases"? If not, I recommend as "CT-negative metastases."

4. It is more intuitive to make the corresponding histogram according to Table 3.

Reviewer #2: According to the results, the authors tell FDG-PET/CT is superior to BS or other modalities as for the highest sensitivity. But it is confined for osteolytic lesion, yes, it's quite lesion-sensitive meaning. Clinically, modality with higher sensitive but lower false positivity when we follow up patients during/after treatment. I think the authors should provide analysis about whether FDG-PET/CT is for osteosclerotic lesion for followup detection. As we know, just benign findings are often reported to osteosclerotic metastasis.

Reviewer #3: Anthors compared lesion-based sensitivity of dual-time-point FDG-PET/CT, bone scintigraphy (BS), and low-dose CT (LDCT) for detection of various types of bone metastases in patients with metastatic breast cancer, and concluded FDG-PET/CT could be considered as a sensitive one-stop-shop in case of clinical suspicion of bone metastases in breast cancer patients. The manuscript is of a certain clinical reference. However, the manuscript needs major revisions.

1.“A total of 488 bone lesions were detected on any of the modalities 35 and were categorized by the LDCT into osteolytic,osteosclerotic,mixed morphologic,or invisible. How to distinguish? from radiologist experience

2. In”Study Design and Subjects” section, author introduce “In a prospective comparative design, we analyzed 100 patients (aged 37-83 years) with 99 suspected recurrent metastatic breast cancer (de-novo), all of whom underwent 100 dual-time-point FDG-PET/CT and whole-body BS, between 2011 (Dec) and 2014 (Sep), 101 within a median time interval of three days (range 0-24). Overall patient-based accuracy 102 results of this study have been published previously (22),” However, the inclusion and exclusion standards of patients in this manuscript are not introduced in detail.

3. The article is used as "1h or 3h FDG-PET/CT" why not “1h or 2h FDG-PET/CT?” What is the reference for this?

4. What is the LDCT Protocol? How to read the imaging by LDCT+ BS，fused？

5. “Three FDG-PET/CT studies did not include the head”. The possible head lesions read by BS，LDCT+ BS, or not？

6. PLOS authors have the option to publish the peer review history of their article (what does this mean?). If published, this will include your full peer review and any attached files.

Reviewer #1: No

Reviewer #2: No

Reviewer #3: No

---

## [Author Response · Author response to Decision Letter 0]

21 Sep 2021

Dear Managing Editor Matteo Bauckneht, dear Editor-in-Chief Emily Chenette.

Thank you for considering the attached manuscript entitled "Diagnosis of Bone Metastases in Breast Cancer: Lesion-Based Sensitivity of Dual-Time-Point FDG-PET/CT Compared to Low Dose CT and Bone Scintigraphy" for revision. We have revised and reformed the manuscript in light of the reviewers’ comments and hope it is now acceptable for publication. The following points indicate how we have addressed the comments and suggestions:

Reply to the comments of reviewer #1:

This manuscript is well-written, however, there are some previous studies have reported the similar results and conclusions, such as Acta Radiol, 2011, 52(9):1009-14., J Clin Oncol,2016, 34(16): 1889-97., Mol Imaging Biol, 2012, 14(2): 252-9., and Mol Imaging Biol, 2020, 22(2): 397-406.

Thanks for your consideration. One of the mentioned references (J Clin Oncol,2016, 34(16): 1889-97) is the parallel study from our research group in which we compared the patient-based sensitivity of modalities regarding detection of bone metastases, while in the current study, we focused on lesion-based sensitivity to complement the previous results. The study by Hahn et al. (Acta Radiol, 2011, 52(9):1009-14), was one of the references of our study and we have added a comparison between the two studies to the discussion section (line 270-273). A combination of FDG PET/CT and NaF-PET/CT addressed in the two other studies for the detection of skeletal metastases, seems to be a different topic; however, this hypothesis is already discussed by senior researchers of our research group (https://pubmed.ncbi.nlm.nih.gov/31808031/).

We believe that our study adds new aspects to the literature since we have considered dual-time-point scanning as well as low-dose computed tomography in a large number of metastatic lesions in a representative group of patients with treatment-naive biopsy-verified metastatic breast cancer, using histopathology as the reference standard.

Please write down the inclusion and exclusion criteria of samples in detail in Study Design and Subjects.

Thanks for mentioning this important point. We have completed the inclusion/exclusion criteria of the study population (methods section, line 98-102).

The scanning range of PET-CT should be performed from the top of the skull to the proximal femur, and metastatic lesions of the skull should not be excluded in the study.

Thanks for mentioning this important point. The upper-thigh field of view including the skull was part of the research protocol of FDG-PET/CT in this study, and we have updated this in the Methods section (FDG-PET/CT protocol, line 118). The seven lesions, located in the skull, were considered in accuracy comparisons between the modalities and were only excluded from quantification analyses (due to scatter from high FDG-uptake in the cerebrum). The exclusion of seven lesions appears negligible in light of the total number of 481 lesions, and their exclusion does not affect the quantification analyses notably.

Is there any other literature supporting the definition of "invisible metastases"? If not, I recommend as "CT-negative metastases."

Thanks for the comment. We have rephrased “invisible metastases” as "CT-negative metastases", as suggested. 

It is more intuitive to make the corresponding histogram according to Table 3.

Thanks for your suggestion. We have provided a histogram chart for SUV corrected for partial volume (cSUV) since it is the most interpretable parameter for clinical practice (Fig 3). 

Reply to the comments of reviewer #2:

According to the results, the authors tell FDG-PET/CT is superior to BS or other modalities as for the highest sensitivity. But it is confined for osteolytic lesion, yes, it's quite lesion-sensitive meaning. Clinically, modality with higher sensitive but lower false positivity when we follow up patients during/after treatment. I think the authors should provide analysis about whether FDG-PET/CT is for osteosclerotic lesion for followup detection. As we know, just benign findings are often reported to osteosclerotic metastasis.

Thanks for the comment. That is true and derived information from follow-up scans could be quite important. However, the patients’ follow-up scans were not part of our research protocol and the patients have been followed-up by different modalities (CE-CT or FDG-PET/CT) based on the decision made in the clinic. However, we used the clinical information within follow-up period (median of 19 months) in labeling the patients as bone-involved malignancies and considered the follow-up information as secondary reference standard. We have added a sentence about the importance of considering follow-up scans in future studies (line 302-304). 

Reply to the comments of the reviewer #3:

Authors compared lesion-based sensitivity of dual-time-point FDG-PET/CT, bone scintigraphy (BS), and low-dose CT (LDCT) for detection of various types of bone metastases in patients with metastatic breast cancer, and concluded FDG-PET/CT could be considered as a sensitive one-stop-shop in case of clinical suspicion of bone metastases in breast cancer patients. The manuscript is of a certain clinical reference. However, the manuscript needs major revisions.

A total of 488 bone lesions were detected on any of the modalities and were categorized by the LDCT into osteolytic, osteosclerotic, mixed morphologic, or invisible. How to distinguish? from radiologist experience

Thanks for the comment. We have added proper reference for categorization of different bone lesions (Image interpretation section, line 152-156). An osteosclerotic lesion has well-defined margins on radiographically findings, while an osteolytic lesion usually appears with a partially ill-defined margin and usually without the development of periosteal reaction. Also, the osteoblastic lesions appear with pattern of bone formation and ossification, while osteolytic lesions appear with bone resorption pattern (focal bone destruction).

In”Study Design and Subjects” section, author introduce “In a prospective comparative design, we analyzed 100 patients (aged 37-83 years) with 99 suspected recurrent metastatic breast cancer (de-novo), all of whom underwent 100 dual-time-point FDG-PET/CT and whole-body BS, between 2011 (Dec) and 2014 (Sep), 101 within a median time interval of three days (range 0-24). Overall patient-based accuracy 102 results of this study have been published previously (22),” However, the inclusion and exclusion standards of patients in this manuscript are not introduced in detail.

Thanks for mentioning this important point. We have completed the inclusion/exclusion criteria of study population (methods section, line 98-102).

The article is used as "1h or 3h FDG-PET/CT" why not “1h or 2h FDG-PET/CT?” What is the reference for this?

Thanks for the comment. You are correct that there are some papers in the literature favoring “1h and 2h FDG-PET/CT” as the protocol of performing dual-time-point FDG-PET/CT. However, there is no definite recommendation according to guidelines on the standard protocol of dual-time-point FDG-PET/CT. Other works focused on delayed scanning 90 minutes (https://pubmed.ncbi.nlm.nih.gov/29439010/) or 100 minutes (https://pubmed.ncbi.nlm.nih.gov/30970638/) after the first injection (mentioned as early delayed scans). We have followed the protocol of 1h and 3h FDG-PET/CT to observe the maximum differentiation of early and delayed scans as earlier suggested in the literature (https://pubmed.ncbi.nlm.nih.gov/21081574/ and https://pubmed.ncbi.nlm.nih.gov/27981471/). We have added respective references to the methods section (FDG-PET/CT protocol, line 119).

What is the LDCT Protocol? How to read the imaging by LDCT+ BS，fused？

The LDCT protocol was added to the methods section (line 127-132). The combined use of LDCT and BS was done following the protocol of having them in two separate screens (side by side) and the radiologist decided on lesions’ categorization.

“Three FDG-PET/CT studies did not include the head”. The possible head lesions read by BS，LDCT+ BS, or not?

Thanks for the comment. We have clarified the issue in the manuscript (line 197-198). Three FDG-PET/CT studies did not include the head by mistake. There was no possibility to check them with LDCT (same field of view with FDG-PET), and BS did not detect any extra lesion within those three patients.

Journal Requirements:

Please ensure that your manuscript meets PLOS ONE's style requirements, including those for file naming.

We have updated all sections according to PLOS ONE's style.

Please provide additional details regarding participant consent. In the ethics statement in the Methods and online submission information, please ensure that you have specified whether consent was informed.

We have already mentioned this point in the first paragraph of the Methods section; written consent form was obtained from all included patients in this study (line 93-97).

We note that the grant information you provided in the ‘Funding Information’ and ‘Financial Disclosure’ sections do not match. When you resubmit, please ensure that you provide the correct grant numbers for the awards you received for your study in the ‘Funding Information’ section.

Thank you for stating the following in the Acknowledgments Section of your manuscript: This study was supported by University of Southern Denmark and Odense University Hospital (Odense, Denmark).

We note that you have provided funding information that is not currently declared in your Funding Statement. However, funding information should not appear in the Acknowledgments section or other areas of your manuscript. We will only publish funding information present in the Funding Statement section of the online submission form. Please remove any funding-related text from the manuscript and let us know how you would like to update your Funding Statement. Currently, your Funding Statement reads as follows: 

This study was supported by University of Southern Denmark and Odense University Hospital (Odense, Denmark). Please include your amended statements within your cover letter; we will change the online submission form on your behalf.

Thanks for the clarification. We have updated the financial disclosure in the Cover letter and removed the related sections from the manuscript.

Please upload a new copy of Figures 1 and 2 as the detail is not clear.

High-resolution format of Figures 1 and 2 are replaced.

Best Regards

Mohammad Naghavi-Behzad (Corresponding Author)

---

## [Editor Report · Decision Letter 1]

3 Nov 2021

Diagnosis of Bone Metastases in Breast Cancer: Lesion-Based Sensitivity of Dual-Time-Point FDG-PET/CT Compared to Low Dose CT and Bone Scintigraphy

PONE-D-20-39838R1

Dear Dr. Naghavi-Behzad,

We’re pleased to inform you that your manuscript has been judged scientifically suitable for publication and will be formally accepted for publication once it meets all outstanding technical requirements.

Kind regards,

Matteo Bauckneht

Academic Editor

PLOS ONE
---

## [Editor Report · Acceptance letter]

8 Nov 2021

PONE-D-20-39838R1 

Diagnosis of Bone Metastases in Breast Cancer: Lesion-Based Sensitivity of Dual-Time-Point FDG-PET/CT Compared to Low Dose CT and Bone Scintigraphy 

Dear Dr. Naghavi-Behzad:

I'm pleased to inform you that your manuscript has been deemed suitable for publication in PLOS ONE. Congratulations! Your manuscript is now with our production department. 

Kind regards, 

on behalf of

Dr. Matteo Bauckneht 

Academic Editor

PLOS ONE